# Outcomes of Patients with Small Intestine Adenocarcinoma in a Canadian Province: A Retrospective Multi-Center Population-Based Cohort Study

**DOI:** 10.3390/cancers14112581

**Published:** 2022-05-24

**Authors:** Emma Yanko, Duc Le, Shazia Mahmood, David Nathan Ginther, Haji Ibraheem Chalchal, Rani Kanthan, Kamal Haider, Adnan Zaidi, Dorie-Anna Dueck, Osama Ahmed, Branawan Gowrishankar, Shahid Ahmed

**Affiliations:** 1College of Medicine, University of Saskatchewan, Saskatoon, SK S7N 4H4, Canada; yanko.emma@usask.ca (E.Y.); duc.le@saskcancer.ca (D.L.); shazia.mahmood@saskcancer.ca (S.M.); n.ginther@usask.ca (D.N.G.); haji.chalchal@saskcancer.ca (H.I.C.); rani.kanthan@saskhealthauthority.ca (R.K.); kamal.haider@saskcancer.ca (K.H.); adnan.zaidi@saskcancer.ca (A.Z.); dorie-anna.dueck@saskcancer.ca (D.-A.D.); osama.ahmed@saskcancer.ca (O.A.); branawan.gowrishankar@saskcancer.ca (B.G.); 2Saskatoon Cancer Centre, University of Saskatchewan, Saskatoon, SK S7N 4H4, Canada; 3Allan Blair Cancer Centre, University of Saskatchewan, Regina, SK S4T 7T1, Canada; 4Department of Surgery, University of Saskatchewan, Saskatoon, SK S7N 5E5, Canada; 5Department of Pathology, University of Saskatchewan, Saskatoon, SK S7N 5E5, Canada

**Keywords:** small intestine adenocarcinoma, small intestine cancer, outcomes, survival, surgery, neutrophil lymphocyte ratio, metastasectomy, performance status, stage

## Abstract

**Simple Summary:**

Small intestine adenocarcinoma is a rare cancer. Given the rarity of this cancer, the clinicopathological characteristics and the role of chemotherapy in both the early and advanced stages have not been well studied. The aim of this study was to assess the outcomes of all patients who were diagnosed with small intestine adenocarcinoma over a 10-year period in the province of Saskatchewan. Most patients with small intestine adenocarcinoma were diagnosed with advanced-stage disease. We found that advanced-stage disease, poor performance status, lack of curative surgery, and a baseline neutrophil:lymphocyte ratio of >4.5 were significantly correlated with inferior survival.

**Abstract:**

Background: Small intestine adenocarcinoma is a rare cancer. The current study aims to determine the outcomes of patients with small intestine adenocarcinoma in a Canadian province. Methods: This retrospective population-based cohort study assessed patients with small intestine adenocarcinoma who were diagnosed from 2008 to 2017 in Saskatchewan. A Cox proportional multivariate regression analysis was performed to determine the correlation between survival and exploratory factors. Results: 112 eligible patients with a median age of 73 years and M:F of 47:53 were identified. Overall, 75% had a comorbid illness, and 45% had a WHO performance status >1. Of the 112 patients, 51 (46%) had early-stage disease and 61 (54%) had advanced-stage disease. The median overall survival (mOS) was as follows: stage one, 59 months; stage two, 30 months; stage three, 20 months; and stage four, 3 months (*p* < 0.001). The median disease-free survival of patients with stage three disease who received adjuvant chemotherapy was 26 months (95% CI:23.1–28.9) vs. 4 months (0.0–9.1) with observation (*p* = 0.04). Patients who received chemotherapy for advanced disease had a mOS of 10 months (3.5–16.5) vs. 2 months (0.45–3.6) without chemotherapy (*p* < 0.001). In the multivariate analysis, stage four disease, hazard ratio (HR), 3.20 (1.84–5.40); WHO performance status >1, HR, 2.22 (1.42–3.45); lack of surgery, HR, 2.10 (1.25–3.50); and a neutrophil:lymphocyte ratio of >4.5, HR, 1.72 (1.10–2.71) were significantly correlated with inferior survival. Conclusions: Most patients with small intestine adenocarcinoma were diagnosed with advanced-stage disease. Advanced-stage disease, poor performance status, lack of surgery and a baseline neutrophil:lymphocyte ratio >4.5 were correlated with inferior survival.

## 1. Introduction

Although the small intestine is a large organ, accounting for about 90% of the mucosal surface of the digestive tract, neoplasms involving the small intestine are rare. Malignant disorders of the small intestine account for less than 5% of all gastrointestinal cancers and about 0.6% of all cancer cases [1,2,3]. A variety of different malignancies arises from the small intestine, such as adenocarcinoma, neuroendocrine tumor, lymphomas, gastrointestinal stromal tumors (GISTs), sarcoma, and melanoma. However, adenocarcinoma and neuroendocrine tumors are the two most common malignancies. Adenocarcinoma accounts for 40% of all small intestine cancers [1,2,3]. Surgical resection and regional lymph node dissection are the primary treatments of localized adenocarcinoma of the small intestine. However, most cases of small intestine cancers are diagnosed in the later stage of the disease when curative surgery is not an option. Because small intestinal adenocarcinoma is a rare malignant neoplasm, its clinicopathological characteristics and the role of chemotherapy in both the early and advanced stages have not been well studied. Given the paucity of prospective data in the management of small intestine cancer, the treatment of small intestine adenocarcinoma has been extrapolated from colon cancer in clinical practice. Most published studies of small intestine adenocarcinoma are retrospective, either from single institutions with an underlying selection bias or having used large registry or administrative datasets, which lacked information on some important clinical variables such as performance status [4,5,6,7,8,9,10,11]. The current study aims to determine the outcomes of patients with small intestine adenocarcinoma who were diagnosed in a Canadian province over a decade.

The study objectives were to determine the overall survival of patients with small intestine adenocarcinoma in relation to the stage and location of the disease, to compare the disease-free and overall survival of patients with stage three disease who received adjuvant chemotherapy to those who did not receive chemotherapy, to compare the overall survival of patients with stage four disease who received palliative chemotherapy to those who did not receive chemotherapy, and to identify the various clinical, pathological, and contextual factors that correlate with overall survival in patients with small intestine adenocarcinoma. 

## 2. Methods

### 2.1. Study Population

This retrospective population-based cohort study involved patients with histologically documented adenocarcinoma of the small intestine who were diagnosed from January 2008 to December 2017 in the province of Saskatchewan, Canada. Patients with another histology, including neuroendocrine tumors, gastrointestinal stromal tumor (GIST), lymphoma, or other active secondary cancers, were excluded. The University of Saskatchewan Biomedical Research Ethics Board approved the study protocol. International Classification of Disease (ICD) codes were used to identify eligible patients using the Saskatchewan Cancer Registry. The Cancer Registry prospectively collects and updates the provincial cancer database. A pre-specified abstraction sheet was used for data collection and one of the researchers (EY) abstracted the individual patient data. 

### 2.2. Definitions

Disease-free survival was defined as the time from surgery until the recurrence of disease, a new primary invasive cancer, or death. Overall survival was defined as the time from diagnosis until death from any cause. Surgical resection was defined as the curative resection of the primary tumor in early-stage disease, or the curative resection of the primary tumor and metastasectomy in patients with advanced-stage disease. Rural residence was defined as per the Statistic Canada definition of Beale non-metropolitan regions that refers to individuals living outside metropolitan regions with urban centers of 50,000 or more in population as rural. Patients were censored at the study cutoff date when they were known to be alive.

### 2.3. Analysis 

Descriptive statistics were performed to summarize the studied population demographics and baseline characteristic parameters. Categorical data were analyzed as the frequency and the corresponding proportion. Fisher’s exact tests and student’s *t*-tests were performed for the analysis of categorical and continuous variables, respectively. The survival of the whole cohort and subgroups was estimated using the Kaplan−Meier method, and the survival distribution of the different groups was compared using the log rank test. The overall significance level was set at 0.05. 

### 2.4. Cox Proportional Regression Analysis

A Cox proportional multivariate regression analysis was performed, and the hazard ratios (HR) and 95% CI were calculated. The following patient-related, tumor-related, and contextual variables were examined with respect to their correlation with an increased mortality: age (<70 vs. ≥70), male gender, World Health Organization (WHO) performance status >1, the presence of a major comorbid illness, rural residence, history of a secondary cancer, history of colorectal cancer, location (duodenum vs. jejunum or ileum), serum creatinine (<120 vs. ≥120 mm/L), albumin (≥35 vs. <35 g/L), alkaline phosphatase (≥140 vs. <140 mm/L), hemoglobin (≥120 vs. <120 g/L), platelets (≥450 vs. <450 × 10^9^/L), median neutrophil to lymphocyte ratio, radiation therapy, surgery (resection of the primary tumor ± metastasectomy), or any chemotherapy. All variables with a *p*-value < 0.20 during univariate analysis were assessed in a multivariate model to assess their relationship with overall survival. The best-fitted but most parsimonious model was developed by identifying the important predictors for survival. The likelihood ratio test and *t* test were used to determine whether the addition of independent variables of interest added significantly to the prediction of survival in the model. A two-sided *p*-value of <0.05 was considered statistically significant. SPSS version 27 was used for statistical analysis (IBM. Armonk, NY, USA).

## 3. Results

### 3.1. Patient Characteristics 

One hundred and twenty-five patients were identified. Of these, thirteen patients were excluded; seven due to another histology, including neuroendocrine tumor, GIST, and lymphoma, five due to limited or no records, and one patient due to metastatic colon cancer with metastases to the small intestine (Figure 1). The median age was 73 years (interquartile range [IQR]: 62–81) and the male:female ratio was 47:53. Among these, 75% had a comorbid illness, 34% had a secondary cancer, and 59% were rural residents. Of the 112 patients, 51 (46%) had early-stage (stage one, two, or three) disease while 61 (54%) had advanced disease. Of the 61 advanced-disease patients, 55 (90%) had a definite stage four disease and 6 (10%) additional patients were suspected of having stage four disease. The patient characteristics are described in Table 1. Significant differences were noted between the patients with early- versus advanced-stage disease in relation to the rate of surgery, delivery of chemotherapy, T4 tumor, node-positive disease, mean low hemoglobin, elevated white blood cell count including neutrophils, and alkaline phosphatase. Most patients had a neutrophil to lymphocyte ratio of greater than 3. The median neutrophil:lymphocyte ratio was 4.5. Overall, 53 (47%) of the patients had duodenal cancer. 

### 3.2. Survival

The median follow-up period was 10.5 months and the total duration of follow up was 150 months. An analysis of survival in relation to stage, performance status, and location showed the significant superior survival of patients with early-stage disease, good performance status, or with jejunal or ileal cancer compared to patients with advanced-stage disease, poor performance status, or duodenal cancer, respectively (Figure 2). The median overall survival in relation to the stage of disease was as follows: stage one, 59 months; stage two, 30 months; stage three, 20 months; and stage four, 3 months (*p* < 0.001). The five-year survival of patients with early-stage disease was 41%; stage one disease was 85%, stage two disease was 44%, and stage three disease was 31%. The overall survival of the entire cohort was 11 months (95% CI: 5.0–17.0). The patients with early-stage disease had a median overall survival of 36 months (95% CI: 8.0–64.0) compared to 3 months (95% CI: 1.74–4.26) for patients with advanced disease (*p* < 0.0001) (Figure 3A).

### 3.3. Early-Stage Disease

Of the 51 patients with early-stage disease, 18 (35%) patients with stage two or three disease received adjuvant or neoadjuvant chemotherapy. Of the 18 patients, 12 (66%) received an oxaliplatin-based treatment (FOLOFX, *n* = 9, CAPOX, *n* = 3) and 6 (33%) patients received 5FU/leucovorin or capecitabine. Overall, 6 patients (12%), mostly with duodenal cancer, received (neo)adjuvant radiation. Of the 51 patients with early-stage disease, 20 (39%) developed recurrent disease, including 18 patients with distant metastases (lung, *n* = 6, liver, *n* = 5, node, *n* = 6, peritoneum, *n* = 10, other sites, *n* = 2). Of the 20 patients with recurrent disease, 14 (70%) received chemotherapy and 6 (30%) underwent resection of the recurrent disease. Of the patients with stage three disease who received adjuvant chemotherapy, their median disease-free survival was 26 months (23.1–28.9) compared to 4 months (0.0–9.1) if they did not receive chemotherapy (*p* = 0.04). Following curative surgery, some patients developed recurrent disease in a short interval, suggesting undetected stage four disease at the time of surgery. The median overall survival of patients with stage three disease who received adjuvant chemotherapy was 47 months (14.8–79.1) compared to 6 months (0.0–16.3) for patients who did not receive adjuvant chemotherapy (*p* = 0.06). 

### 3.4. Advanced-Stage Disease 

Of the 112 patients investigated, 39 received chemotherapy for recurrent or advanced disease, and 25 (41%) of the 61 patients with advanced disease received chemotherapy. Of the 61 patients with advanced disease, 26 (42%) underwent surgery, including 22 patients who underwent resection of the primary tumor with or without the resection of metastatic disease. Patients who received chemotherapy for advanced disease had a median overall survival of 10 months (3.5–16.5) compared to 2 months (0.45–3.6) if they did not receive chemotherapy (*p* < 0.001) (Figure 3B). Patients with stage four disease who received chemotherapy and underwent surgery had a median overall survival of 18 months (13.7–23.3). 

### 3.5. Cox Proportional Regression Analysis

For the univariate analysis, the following characteristics were significantly correlated with inferior survival: aged 70 years or older, WHO performance status > 1, stage four disease, low albumin, elevated creatinine, high alkaline phosphatase, duodenal cancer, neutrophil:lymphocyte ratio > 4.50, lack of chemotherapy, and absence of surgery (Table 2). Among them, stage four disease was most strongly correlated with high mortality, with a hazard ratio (HR) for death of 4.32 (95% CI: 2.7–7.0). After adjustment for other significant variables, the most parsimonious model revealed that stage four disease, HR, 3.20 (1.84–5.40); WHO performance status of >1, HR, 2.22 (1.42–3.45); lack of curative surgery, HR, 2.10 (1.25–3.50); and a neutrophil:lymphocyte ratio of >4.5, HR, 1.72 (1.10–2.71) were significantly correlated with inferior overall survival in patients with small intestine adenocarcinoma (Table 3).

## 4. Discussion

Few studies to date have evaluated the effect of various clinical and pathological variables on the overall survival of patients with small intestinal adenocarcinoma. The current study shows that, in addition to well-known variables including the stage of disease and performance status at the time of diagnosis, a lack of surgery with a curative intention and a baseline neutrophil:lymphocyte ratio value of >4.5 correlate with an inferior overall survival in patients with small intestine adenocarcinoma. 

Surgery is the mainstay of treatment for localized adenocarcinoma of the small intestine. Numerous studies have shown that if surgical resection is feasible, the goal should be margin-negative resection (R0); detailed surgical treatment depends on the location of the primary lesion and disease stage [12,13]. However, there is uncertainty about the benefit of metastasectomy in patients with advanced small intestine cancer. A large single institution study from MD Anderson showed that metastasectomy was associated with a better overall survival in well-selected patients with advanced small intestine adenocarcinoma [14]. Overall, 17% of patients with advanced disease underwent resection of the primary tumor and metastases. Of note, following multivariate analysis, metastasectomy was not shown to be independently associated with better survival. Another small cohort study involving 34 patients treated with the resection of metastases showed that the overall survival of patients with advanced small intestine adenocarcinoma was poor, despite metastasectomy; however, long-term survival was observed in some patients [15]. A multicenter study from Japan reported that approximately 18% of patients with advanced small intestine adenocarcinoma who were treated with a combination of resection of the primary tumor, metastasectomy, and chemotherapy had a median overall survival of around 36 months [16]. In our study cohort, 14% patients with advanced small intestine cancer underwent metastasectomy. The patients who had metastasectomy and received chemotherapy had a significantly longer survival compared to patients who received chemotherapy alone. In contrast, about 20% of patients with localized small intestine cancer (mostly duodenal cancer) did not undergo curative surgery due to their unsuitability for radical surgery. After adjustment for all other known and exploratory clinical variables, patients who did not have resection of the primary tumor for early-stage disease or primary tumor resection and metastasectomy for advanced-stage disease had poor survival compared to those who underwent surgery. Our findings support previous observations that regardless of disease stage, if surgery with curative intention is feasible, it is associated with a better outcome. It is important to note that the median age of the study cohort was 73 years, which is higher than that of other reports. Age is an important prognostic characteristic and correlates with an inferior overall survival in cancer patients. Howe et al. reported on patients aged > 75 years with small intestine cancer; their relative risk of death was 1.8 times higher, which may be explained by the fact that (a) tumors located in the duodenum are more common in this group, and (b) surgical resection is performed less often [17]. 

Few baseline laboratory values in patients with small intestine adenocarcinoma have been shown to be significantly correlated with a decreased overall survival. For example, Sakae et al. reported that a baseline high CEA, high LDH, and low albumin were correlated with inferior survival [16]. In our study cohort, a neutrophil:lymphocyte ratio of >4.5, low albumin, high creatinine, and high alkaline phosphatase were associated with an increased risk of death. Since CEA and LDH were not routinely performed, these variables were not examined in the multivariate model. However, after adjustment for other variables, only a baseline neutrophil:lymphocyte ratio of >4.5 was correlated with poor overall survival. The neutrophil:lymphocyte ratio has been established as a biomarker of poor prognosis in several cancers, including gastric cancer and colorectal cancer [18,19,20]. A study by Vano et al. reported that the neutrophil:lymphocyte ratio is associated with poor prognosis independent of tumor type [21]. The same study reported that the “optimal” pre-treatment cut-off in patients with solid tumors is between 3.5 and 4.5. Given a high baseline neutrophil:lymphocyte ratio value in our study cohort, we used the median value of 4.5 for multivariate analysis. Vano et al. suggest that monitoring the variation in the neutrophil:lymphocyte ratio during systemic therapy may predict the response to treatment and prognosis. They reported that a standard deviation increase of one of the neutrophil:lymphocyte ratio from baseline was correlated with a 35% increased risk of death. The current study used a median neutrophil:lymphocyte ratio cut-off of 4.5 and showed that patients with small intestine adenocarcinoma, regardless of disease stage, with a neutrophil:lymphocyte ratio of >4.5 experienced an approximately two-fold increased risk of death. A high neutrophil:lymphocyte ratio is a surrogate maker of underlying systemic inflammatory processes that may contribute to the angiogenesis, growth, invasion, and metastasis of tumor cells and poor outcomes [22,23,24]. Further research is needed to determine whether variation in the neutrophil:lymphocyte ratio in patients with small intestine adenocarcinoma is a better prognostic marker.

Previous studies have reported the duodenum to be the most common site of small intestine adenocarcinoma, complementing the findings of the current study [2,12,13]. Typically, the second and third most prevalent locations are the jejunum and ileum [2,12,13]. Duodenal adenocarcinoma has been previously reported to correlate with a significant decrease in 5-year survival and median overall survival when compared to jejunal or ileal tumors [2,12,25]. Our results also showed that patients with duodenal cancer had a 73% lower overall survival compared to the other sites; however, after adjustment for other variables, the location of small adenocarcinoma was not independently correlated with survival.

Disease stage and performance status are well understood prognostic factors for many cancers. However, due to the rarity of small intestine adenocarcinoma, the correlation between patients’ WHO performance status and overall survival is not well reported. The current study demonstrated that a WHO performance status > 1, regardless of the stage of the disease, was associated with a more than two-fold increased risk of death in patients with small intestine adenocarcinoma. A previous study similarly reported that an Eastern Cooperative Oncology Group (ECOG) performance status of 3–4 in all stages of small intestine adenocarcinoma was associated with poor overall survival [16]. Likewise, most of the patients included in this study were diagnosed with stage four disease. The overall survival of patients with advanced-stage disease was significantly lower in comparison to those with early-stage disease. It is important to note that most patients with advanced-stage disease were older with a poor performance status and did not receive palliative chemotherapy. In fact, about 50% of patients died within 3 months of diagnosis. 

Previous studies have reported a benefit of palliative chemotherapy treatment in patients with advanced small intestine adenocarcinoma [14,16,26,27,28,29]. However, the benefit of chemotherapy treatment in early-stage disease (stage one–three) remains unclear [26,30]. In the current study, the minority of patients with early- and advanced-stage disease received chemotherapy: 35% and 41%, respectively. Oxaliplatin-based chemotherapy was most commonly used. Some studies have suggested that overall survival is significantly improved in patients with stage three disease who receive adjuvant chemotherapy [31,32]. Conversely, other studies did not confirm the benefit of adjuvant chemotherapy [26,33,34]. The current study demonstrates a significant increase in the overall survival of patients with advanced cancer who received chemotherapy. Likewise, an improvement in disease-free survival and numerical increases in the overall survival of patients with stage three disease who received adjuvant chemotherapy was noted. However, upon multivariate analysis, when chemotherapy was examined as a variable in all stages, it was not associated with a better overall survival. Of note, patients with stage three disease who did not have adjuvant chemotherapy had a much shorter disease-free survival and overall survival. Some patients developed recurrent disease in the short interval after surgery, suggesting that they may have had undetected stage four disease at the time of surgery. The paucity of information regarding adjuvant chemotherapy use for early-stage small intestine adenocarcinoma is in part due to the lack of randomized clinical trials and requires prompt attention [35].

The key limitations of the current study include a relatively small sample size and the retrospective nature of the study. In addition, information on biomarkers such as microsatellite instability status, was not available for the majority of patients. Nevertheless, the key strengths of the current study include the absence of selection bias and, despite having a small sample size in comparison to other population-based studies, we have included more detailed information about individual patients, such as their baseline performance status. 

## 5. Conclusions

In keeping with previous studies, adenocarcinoma is an aggressive malignancy. Most patients with small intestine adenocarcinoma were diagnosed with advanced-stage disease. In addition to advanced-stage disease and poor performance status, a lack of curative surgery and a baseline neutrophil:lymphocyte ratio of >4.5 were significantly correlated with inferior survival. Future prospective studies will be helpful to confirm the findings.

## Figures and Tables

**Figure 1 cancers-14-02581-f001:**
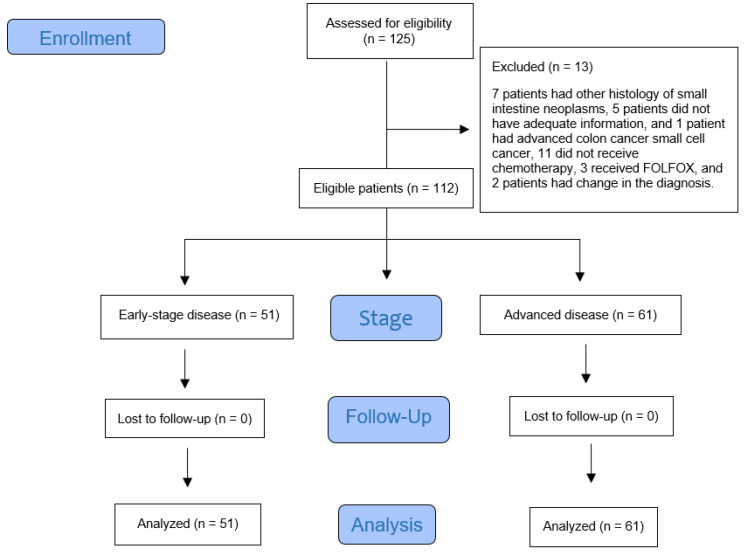
Flow diagram of eligible patients with small intestine adenocarcinoma cancer who were treated with two different combination chemotherapy regimens during the study period.

**Figure 2 cancers-14-02581-f002:**
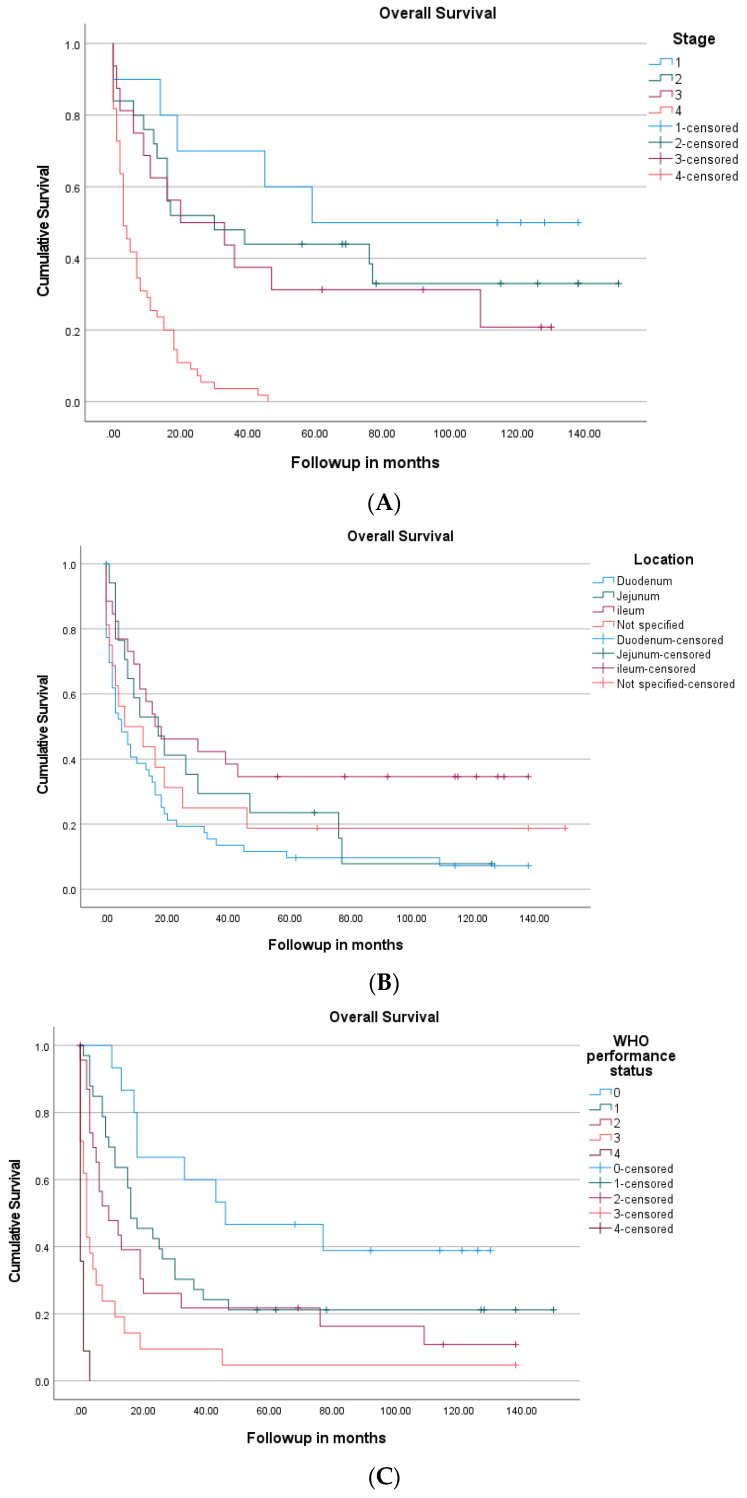
Kaplan–Meier survival curves of patients with small intestine cancer in relation to stage (**A**), location (**B**), and WHO performance status (**C**).

**Figure 3 cancers-14-02581-f003:**
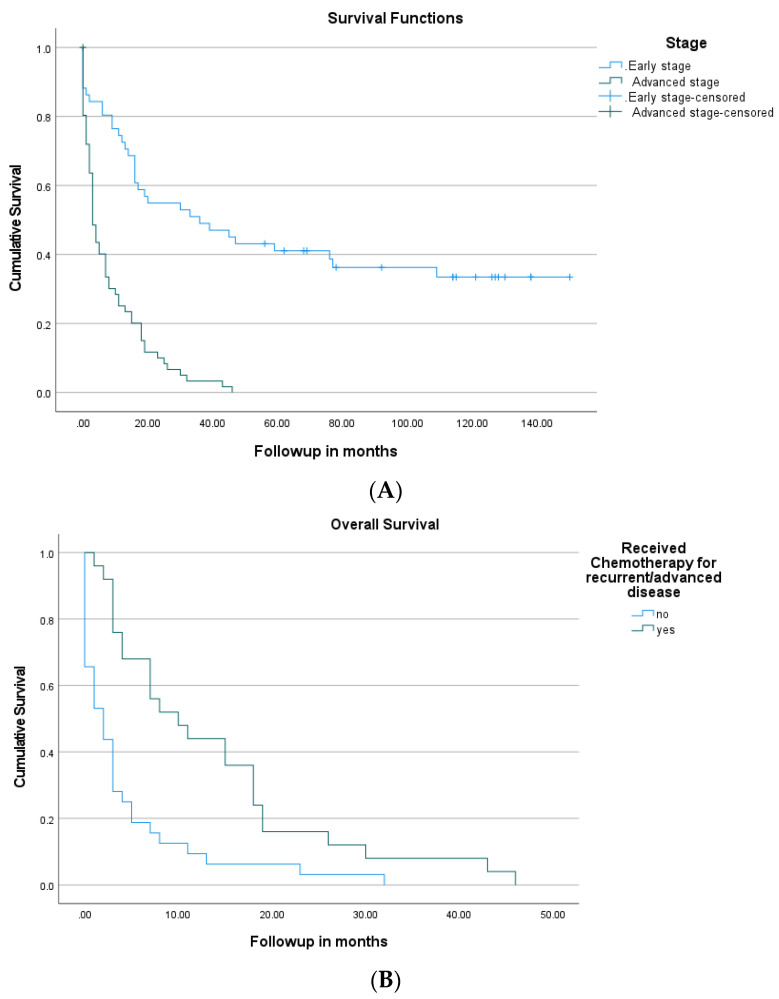
(**A**): Comparison of Kaplan–Meier survival curves of patients with early-stage disease to those with advanced disease. (**B**): Comparison of Kaplan–Meier survival curves of patients with advanced-stage small intestine adenocarcinoma who were treated with chemotherapy vs. no chemotherapy.

**Table 1 cancers-14-02581-t001:** Baseline characteristics of the entire cohort and subgroups of patients with early stage (stage I, II, and III) and advanced disease (stage IV or unknown) with small intestine adenocarcinoma.

Variables	All Patients N = 112 (%)	Early Stage Disease N = 51 (%)	Advanced Disease N = 61 (%)	*p* Value
Median age in years	73 (IQR: 62–81)	71 (IQR: 62–79)	75 (IQR: 62–82)	0.15
Men	53 (47)	27 (53)	26 (43)	0.34
Rural resident	66 (59)	29 (57)	37 (61)	0.70
Comorbid illness *	74 of 99 (75)	37 of 46 (80)	37 of 53 (70)	0.25
Secondary cancer *	35 of 103 (34)	15 of 48 (31)	20 of 55 (36)	0.67
WHO performance status < 1	48 (45)	26 (55)	22 (37)	0.08
Location				
Duodenum	53 (47)	20 (39)	33 (54)	0.13
Jejunum	17 (15)	9(18)	8 (13)	0.60
Ileum	26 (23)	15 (29)	11 (18)	0.18
Not specified	16 (14)	7 (14)	9 (15)	1.0
Resection of primary tumor ± metastases	63 (56)	41 (80)	22 (36)	0.0001
Resection of metastases	9 (14)	6 (15)	3 (14)	1.0
T4 tumor	50 (45)	20 (39)	30 (49)	0.0005
Node negative disease	53 (47)	32 (63)	21 (34)	0.015
Stage				
I	10 (9)	10 (20)	-	
II	25 (22)	25 (49)	-	
III	16 (14)	16 (31)	-	
IV	55 (49)	-	55 (90)	
Not known **	6 (5)	0	6 (10)	
Mean creatinine	91.4 ± 75.3	97.3 ± 91.6	86.1 ± 58	0.50
Mean albumin	31.5 ± 7.7	33.1 ± 6.2	30.1 ± 8.6	0.11
Mean alkaline phosphatase	164 ± 153.4	102 ± 92.5	214 ± 173	0.001
Mean WBC	8.9 ± 3.6	8.0 ± 3.3	9.7 ± 3.6	0.02
Mean hemoglobin	113 ± 22	120 ± 22	108 ± 22	0.016
Mean platelets	310 ± 132	299 ± 111	320 ± 150	0.48
Mean lymphocytes	1.38 ± 0.71	1.39 ± 0.60	1.38 ± 0.81	0.95
Mean neutrophils	6.40 ± 3.32	5.54 ± 3.22	7.14 ± 3.18	0.03
Mean neutrophil: lymphocyte	6.1 ± 6.1	5.8 ± 8.0	6.3 ± 3.8	0.71
Chemotherapy				
(neo)Adjuvant	20 (18)	18 (35)	2 (3)	<0.0001
Recurrent/metastatic disease	39 (35)	14 (28)	25 (41)	0.17
Received radiation	14 (13)	6 (12)	8 (13)	1.0
Palliative	8 (57)	2 (33)	6 (75)	0.28

* Information was missing in some patients; ** based on their clinical presentation stage IV disease was suspected by treating physicians; ± standard deviation; WHO: World Health Organization.

**Table 2 cancers-14-02581-t002:** Cox regression univariate analyses for assessment of various variables and overall survival of patients with small intestine cancer.

Variables	HR 95% CI	*p*
Age ≥ 70 years	1.66 (1.07–2.57)	0.02
Male sex	1.35 (0.90–2.04)	0.15
Comorbid illness	0.92 (0.57–1.51)	0.75
Secondary cancer	1.10 (0.70–1.72)	0.68
Rural residence	1.04 (0.69–1.58)	0.84
H/O colorectal cancer	0.90 (0.53–1.55)	0.90
WHO performance status > 1	2.50 (1.62–3.81)	<0.001
Stage 4 disease	4.32 (2.70–7.0)	<0.001
Albumin < 35 g/L	2.0 (1.20–3.38)	0.01
Creatinine > 120	1.70 (1.11–2.62)	0.015
Alkaline phosphatase > 140	2.38 (1.56–3.61)	<0.001
Hemoglobin < 120 g/L	1.22 (0.79–1.90)	0.36
Platelets > 450	1.65 (0.82–3.30)	0.18
Duodenum	1.73 (1.15–2.62)	0.008
Neutrophil:lymphocyte ratio > 4.5	2.26 (1.44–3.55)	<0.001
No chemotherapy	1.45 (0.95–2.20)	0.080
No primary tumor resection ± metastasectomy	4.0 (2.51–6.38)	<0.001
Radiation therapy	1.08 (0.61–1.91)	0.78

**Table 3 cancers-14-02581-t003:** Cox regression multivariate analyses and modelling of various clinical and pathological variables and their relationship with overall survival.

Variables	HR (95% CI)	*p*	HR (95%CI)	*p* Value
Age ≥ 70 years	1.31 (0.78–2.22)	0.31		
Age < 70 years	1			
Men Women	1.26 (0.77–2.10) 1	0.35		
WHO performance status > 1	2.01 (1.17–3.46)	0.012	2.22 (1.42–3.45)	<0.001
WHO performance status ≤ 1	1			
Stage 4 disease	3.0 (1.74–5.16)	<0.001	3.20 (1.84–5.40)	<0.001
Stage 1, 2, or 3 disease	1			
Albumin < 35 g/L	1.05 (0.56–1.97)	0.87		
Albumin ≥ 35 g/L	1			
Creatinine > 120	1.42 (0.76–2.66)	0.27		
Creatinine ≤ 120	1			
Alkaline phosphatase > 140	1.47 (0.78–2.77)	0.23		
Alkaline phosphatase ≤ 140	1			
Duodenum	1.20 (0.68–2.10)	0.53		
Jejunum, ileum, or not known	1			
Neutrophil:lymphocyte ratio > 4.5	1.90 (1.10–3.28)	0.02	1.72 (1.10–2.71)	0.019
Neutrophil:lymphocyte ratio ≤ 4.5	1			
No chemotherapy	1.02 (0.56–1.84)	0.96		
Received chemotherapy	1			
No surgery	1.96 (1.02–3.80)	0.04	2.10 (1.25–3.50)	0.005
Surgery	1			

## Data Availability

The data presented in this study are not publicly available. Data access will require approval form the University of Saskatchewan Biomedical Ethics Board and Data Access Committee of the Saskatchewan Cancer Agency.

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
