# Peer review of "Outcomes of Patients with Small Intestine Adenocarcinoma in a Canadian Province: A Retrospective Multi-Center Population-Based Cohort Study"

_cancers, 2022, doi:10.3390/cancers14112581_

Round 1
Reviewer 1 Report
Dear authors,
Thank you very much for reviewing, clarifying and modifying the allegations made previously. It is a very constructive work.
Kind regards
Reviewer 2 Report
The authors addressed all of my comments
This manuscript is a resubmission of an earlier submission. The following is a list of the peer review reports and author responses from that submission.
Round 1
Reviewer 1 Report
Dear Authors:
The authors have carried out the study “Outcomes of Patients with Small Intestine Adenocarcinoma in a Canadian Province. It is a retrospective population-based cohort study involving 112 patients with histologically documented adenocarcinoma of the small intestine diagnosed between 2008-2017.
The study is well designed and rigorous and aims to determine the outcomes of patients with small intestine adenocarcinoma who were diagnosed in a Canadian province over a decade.
From an academic point of view the manuscript gives a good general knowledge of the natural history of patients with small intestine adenocarcinoma and its usual evolution in a specific geographic área.
Some considerations need to be taken into account:
- The small sample of the study reduces the power of the conclusions, although the data collection is profuse.
- Why did the authors collect only adenocarcinomas instead of collecting all histological subtypes of malignant tumors of the small intestine? That analysis would have been very enriching and with a much more generous sample size.
- The authors do not mention the limitations of their study, especially as it is a retrospective study.
- It would be advisable to update the bibliographical referencesIn conclusion, it is a well designed study that needs minor revision and in my opinión, is suitable for publication if you consider so.
- Although the topic is not new, the data collection is extensive and gives us a very approximate idea of the clinical scenario under study.
Kind regards
Reviewer 2 Report
Dear authors,
I’ve read carefully your manuscript. The analysis is well performed, and the topic is interesting indeed.
Overall, this retrospective analysis, despite some interesting insights, is limited by some important issues.
First of all, the number of included patients is too low, compared to other retrospective series recently published. Other retrospective series on small bowel adenocarcinoma have not been cited in the text (i.e.: PMID 34972663, 34950695, 34263641, 33113018). The period “Most published studies of small intestine adenocarcinoma are retrospective, either from single institutions with underlying selection bias or having used large registry or administrative datasets, which lacked information on important clinical variables” is not supported by adequate references.
Second, data about mismatch repair deficiency and the potential impact of Lynch syndrome on survival and diagnosis have not been reported.
Third, the metastatic pattern has not been reported by the authors (please see PMID 34950695); the low number of patients who underwent metastasectomy makes it difficult to assess the role of this kind of surgery. There are other works on this topic, i.e. PMID: 33940348
Fourth, the median NLR is higher than what observed in other cancer types: authors should report also median value of neutrophils and median value of lymphocytes observed in the cohort. Other scores, such as the platelet-lymphocyte ratio (PLR), should be investigated.
Minor comment: the ratio M:F (53:59) is wrong, did you mean 47:53?
Reviewer 3 Report
Yanko et al. performed a retrospective study, in which they analyzed small intestine adenocarcinoma in a Canadian province . According to their results, most patients with small intestine adenocarcinoma were diagnosed with advanced-stage disease. Moreover, advanced-stage disease, poor performance status, lack of surgery and baseline neutrophil:lymphocyte ratio >4.5 were correlated with inferior survival.
The study is well-structured and well-written. Moreover, small intestine carcinoma is a rare identity, which makes the study much more interesting. I have no further recommendations.